# AI Progress Should Be Measured by Capability-Per-Resource, Not Scale Alone: A Framework for Gradient-Guided Resource Allocation in LLMs

**David McCoy** [*] [§]          **Yulun Wu** [†] [§]          **Zachary Butzin-Dozier** [§]

## Abstract

This position paper challenges the "scaling fundamentalism" dominating AI research, where unbounded growth in model size and computation has led to unsustainable environmental impacts and widening resource inequality. We argue that LLM development should be fundamentally reoriented toward capability-per-resource rather than capability alone. We present a theoretical framework demonstrating that resource-allocation decisions guided by gradient influence patterns can dramatically improve efficiency throughout the AI lifecycle. Our analysis shows that in transformer-based models, where a small fraction of parameters exert outsized influence (following heavy-tailed distributions), three critical insights emerge: (1) updating only high-influence parameters strictly outperforms full-parameter tuning on a performance-per-resource basis; (2) simple gradient norms provide computationally efficient proxies for identifying these high-influence components; and (3) coordinated parameter and data selection yields multiplicative efficiency gains, potentially reducing resource requirements by orders of magnitude. Building on these theoretical foundations, we propose a two-stage paradigm—marginal-return pretraining for foundation developers and influence-guided adaptation for downstream users—bridged by gradient blueprints, metadata describing which parameters matter most for various tasks. This capability-per-resource perspective transforms what were once considered pragmatic hardware workarounds into theoretically optimal strategies, democratizing access to cutting-edge AI capabilities while significantly reducing environmental impact. By embedding resource consciousness into how we develop, adapt, and evaluate models, we can reshape AI progress toward a more sustainable and equitable future.

## 1 Introduction

Training a single large language model can demand extraordinary resources: GPT-3 produced 552 tons of $CO_2$ equivalent [26], while LLaMA 65B demonstrated striking diminishing returns with its final 0.2T tokens yielding less than 0.01 improvement in validation loss despite consuming approximately 15% of total training compute [31]. This "scaling fundamentalism" – the pursuit of ever-larger models without explicit consideration of resource efficiency – has both environmental consequences [29, 2] and exacerbates the divide between well-resourced industry labs and the broader research community [6, 5].

**LLM development should be guided by capability-per-resource rather than capability alone, with resource-allocation decisions driven by gradient influence patterns at both parameter and data levels.**

§ Division of Biostatistics, University of California, Berkeley
† Capital One
* Correspondence to: `david_mccoy@berkeley.edu`

39th Conference on Neural Information Processing Systems (NeurIPS 2025) Position Paper Track.

Today's AI landscape consists of two distinct tiers: foundation model developers with massive computational resources, and downstream adapters working under strict resource constraints [6]. While scaling laws [17, 13] guide initial architecture selection, they don't address two critical efficiency questions: (1) when foundation developers should halt training as returns diminish, and (2) which specific parameters downstream users should prioritize when adapting these models with limited resources. These decisions currently lack formal resource-conscious frameworks despite growing calls for systematic efficiency reporting [12, 22].

Our primary contribution is a formal theory of **gradient-guided resource allocation** showing that focusing on high-influence components can dramatically improve efficiency in LLM development. This theory operates along two critical dimensions: **1. Parameter-wise efficiency**: We prove that under realistic power-law gradient distributions, updating only the most influential parameters can strictly outperform full-parameter tuning on a performance-per-resource basis. This provides theoretical foundations for the empirical success of methods like LoRA [15] and QLoRA [8]. **2. Data-wise efficiency**: The same theoretical framework extends to training data, where we demonstrate that not all examples contribute equally to learning [18, 30]. By identifying high-gradient training examples that maximally impact relevant parameters, we can achieve further multiplicative efficiency gains.

Building on this unified framework, we introduce **gradient blueprints**—metadata released alongside model weights that reveal which parameters and data patterns matter most. For example, one blueprint might identify that for English-language understanding tasks, 13% of mid-layer attention parameters carry 85% of the total gradient influence, enabling targeted fine-tuning on just those components. In a multilingual translation scenario, the blueprint might show that cross-attention blocks in later layers contribute disproportionately to performance, whereas for math-heavy reasoning tasks it highlights feed-forward submodules near the final layers. By selectively tuning only these high-influence subsets, downstream practitioners can adapt the model at a fraction of the cost without forfeiting accuracy. Our formal analysis demonstrates three key results: (1) gradient distributions in transformer models follow power-law patterns [24, 21] where a small fraction of parameters exert outsized influence; (2) simple gradient norms effectively approximate more complex influence measures without requiring second-order computations; and (3) memory savings from selective parameter updates enable practical efficiency gains by reducing optimizer state storage requirements, which are proportional to the number of trainable parameters. This holistic resource-efficiency approach enables the combination of selective parameter updates and targeted data selection to achieve significantly higher performance-per-resource than conventional approaches, while democratizing access to advanced AI capabilities across a broader range of institutions and researchers.

In §2, we discuss related work in parameter-efficient fine-tuning and resource tracking. In §3, we introduce our resource-conscious framework, with emphasis on gradient blueprints for downstream adapters. §4 presents theoretical foundations showing why partial updates can outperform full-parameter tuning. §5 outlines practical implementations of gradient-centric model releases. Finally, §6 discusses broader implications and future directions. *Our position challenges the dominant paradigm that ever-larger models and more computation will inevitably lead to better AI*. Proponents of scaling argue that: (1) increasing model size has reliably produced qualitative leaps in capability [17, 33]; (2) resource constraints are temporary and will be solved through hardware innovation; and (3) focusing on efficiency might slow progress toward important breakthroughs. These views have merit – scaling has indeed driven remarkable advances. However, they overlook several key factors: (1) the environmental impact of unbounded scaling is substantial and growing [29, 23]; (2) hardware efficiency gains have not kept pace with model size growth; and (3) the concentration of AI capabilities among well-resourced labs threatens broader scientific progress [1, 25]. Most importantly, even scaling advocates recognize that simply training larger models isn't optimal – [13]'s (2022) Chinchilla scaling laws demonstrate that resource allocation between model size and training compute matters. Our position does not oppose scaling entirely, but rather advocates for embedding resource-consciousness within scaling decisions. By making capability-per-resource the north star metric, we can continue AI advancement while ensuring sustainability, democratization, and optimal resource allocation.

## 2    Related Work: The Efficiency Divide in AI Development

**Scaling Laws and Resource Constraints.**  AI progress has been dominated by what we term *scaling fundamentalism*—the belief that increasing model size and training data will inevitably yield superior capabilities [17, 13].  While the "Chinchilla" scaling laws [13] provide valuable insights into optimal compute allocation, they fundamentally operate *within* fixed computational budgets, treating environmental costs as external constraints rather than intrinsic optimization targets. This has produced a research landscape where performance improvements eclipse environmental considerations, despite growing evidence of AI's substantial carbon footprint [29, 26].

The resulting computational demands have created what Bender et al. [2] call a "danger of stochastic parrots"—increasingly capable models that consume disproportionate resources while amplifying biases and excluding marginalized voices. This pattern reinforces a stark divide in the AI ecosystem: a handful of well-resourced institutions develop massive foundation models, while the broader research community must operate under strict resource constraints [6]. This concentration of computational power has produced what some researchers describe as "computational oligarchs" [5], driving a monoculture where resource-rich organizations dictate global AI priorities [25, 1].

**Parameter and Data Efficiency Approaches.** The resource constraints faced by most researchers have spurred development of various efficiency-enhancing techniques. Parameter-efficient fine-tuning methods like LoRA [15], QLoRA [8], and adapters [14] reduce memory requirements by updating only a small subset of parameters [10, 9, 11].  Post-training approaches such as pruning [9, 11] and attention head elimination [24] remove unnecessary components after full training [27]. These methods effectively reduce inference and adaptation costs but are typically presented as pragmatic workarounds rather than principled optimization strategies.  Parallel efforts have addressed data efficiency through importance sampling [18], curriculum learning [3], and coreset selection [28]. Recent empirical investigations have revealed heavy-tailed gradient distributions in transformers [24, 21], suggesting that both parameters and data points exhibit highly skewed influence distributions. However, these approaches have developed largely independently, without a unified theoretical framework that spans both parameter and data dimensions.

**Evaluation and Reporting Frameworks.** Recent initiatives have begun to address resource efficiency in AI evaluation. The Holistic Evaluation of Language Models (HELM) framework [22] incorporates efficiency metrics such as training cost per kWh or $CO_2$ emitted, but primarily serves as an evaluation tool rather than offering mechanisms to systematically improve efficiency. Similarly, environmental impact statements [23] and carbon accounting systems [12, 20] promote transparency but do not provide concrete guidance for resource optimization during model development and adaptation. Patterson et al. [26] demonstrated that architectural choices can reduce carbon footprint by 100–1000$\times$, but stopped short of providing a formal framework for identifying which specific parameters matter most.  This reveals a critical gap in the literature: despite growing awareness of resource constraints, there exists no principled framework that unifies parameter and data efficiency, provides theoretical guarantees for selective updates, and offers practical mechanisms for efficient knowledge transfer between foundation model developers and downstream adapters.

## 3    A Principled Framework for Resource-Conscious LLM Development

**From Capability to Capability-Per-Resource.**  We propose measuring progress by how much performance $\Psi$ improves *per unit of resource* $\Gamma$ spent. This is complementary to scaling laws: scaling helps choose the size–data regime in advance, while capability-per-resource guides decisions *during* training about when to stop and what to update. Concretely, we stop when the ratio $\Delta\Psi/\Delta\Gamma$ *averaged over a short moving window* falls below a threshold $\eta$ and stays below it for $P$ successive checks (a simple "patience" rule that handles non-monotonic learning curves). In practice, report: (i) the task score $\Psi$; (ii) the total resource $\Gamma$ (e.g., GPU-hours, energy in kWh, or VRAM-GB$\times$hours) and, when possible, its carbon estimate; and (iii) the final $(\Psi, \Gamma)$ pair and the last-window ratio.

**Two-Stage Resource-Conscious Approach.** Our framework addresses two distinct communities in the "bifurcated AI ecosystem": *(1) Foundation-model developers*, who have abundant compute and train general-purpose models, and *(2) Model adapters (or users)*, who specialize these checkpoints for narrower domains under stricter budgets through fine-tuning.

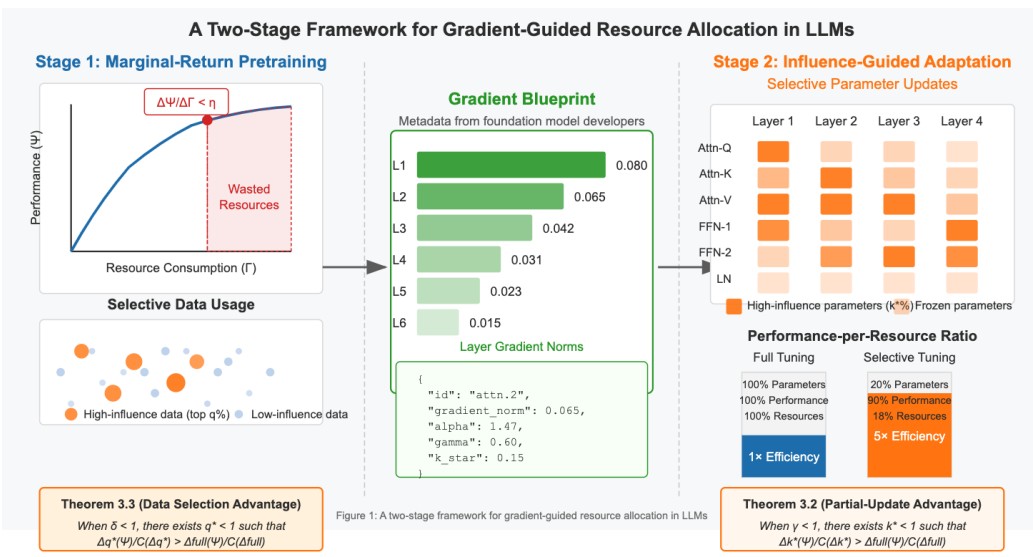

Figure 1: **Resource-Conscious LLM Lifecycle.** (*Left*) Stage 1 halts pretraining once $\Delta\Psi/\Delta\Gamma$ dips below $\eta$. (*Right*) Stage 2 fine-tunes only high-influence submodules (selected via blueprint data), improving performance-per-resource.

*Stage 1: Marginal-return pretraining (foundation labs).* Within a chosen scaling regime, track $\Delta\Psi/\Delta\Gamma$ over short moving windows and stop when the averaged ratio stays below $\eta$ for $P$ successive checks. *Scope:* this applies to foundation labs already running large pretraining; downstream users are not expected to pretrain. $\Delta\Gamma$ can grow late in training due to optimizer/state footprint and I/O (e.g., checkpointing) [34, 26]. Making these costs explicit can save double-digit percent compute at negligible loss [31, 13]. *Stage 2: Influence-Guided Adaptation (Model Adapters).* Once a pretraining checkpoint is released, downstream practitioners need only fine-tune a *fraction* of parameters to achieve strong performance—provided they know *which* parameters matter most. To guide this, we propose foundation developers publish *gradient blueprints*: metadata summarizing submodule-level gradient norms on representative tasks. Adapters then update only those *high-influence parameters* (e.g. 10–20% of total), thereby reducing memory overhead and enabling more epochs or larger batch sizes within a fixed budget.

**Cross-Influence and Multiplicative Gains.** In addition to *parameter* selection, we propose selective *data* usage to further amplify efficiency. We introduce a *cross-influence tensor* that quantifies the interaction between specific parameters and specific training examples. Selecting only high-influence parameter-data pairs yields *multiplicative* savings: for instance, if parameter pruning retains 80% performance using 20% of parameters, and data filtering retains 90% performance using 30% of data, their combination can retain 72% of performance at only 6% of total resource cost.

**Implementation and Real-World Considerations.** *(1) Automated Blueprint Tools:* We envision open-source packages to log submodule-level gradients during validation, output them in standardized JSON format, and guide partial updates downstream. *(2) Gradient Prediction Networks:* To avoid processing all training data just to find "high-influence examples," lightweight networks can estimate gradient norms for unprocessed samples. *(3) Public Releases of Blueprints:* Foundation labs can publish gradient norms or approximate "influence maps" for each submodule, letting adapters quickly identify top-$k\%$ parameters. *(4) Protocol for Adaptation:* When facing a new domain (e.g., medical text), users can blend blueprint statistics with local gradient checks on a small in-domain sample, refining the set of parameters they update.

**Why Resource Awareness Matters.** By embedding explicit resource metrics into both pretraining and adaptation, we transform open-ended scaling into a principled search for the highest-value use of compute. Rather than treating efficiency as an afterthought, our framework *prioritizes* it from the outset, enabling smaller labs to access state-of-the-art performance without massive budgets, and reducing environmental impact by potentially orders of magnitude. In short, we reconceive "bigger is

better" as "better per unit resource," democratizing advanced AI capabilities through a fundamentally more sustainable paradigm.

# 4 Theoretical Foundations of Resource-Conscious Training

We now present the mathematical underpinnings of our approach. In §4.1, we show that under realistic heavy-tailed gradient distributions, *updating only a fraction of parameters* can be more cost-effective than tuning them all. In §4.2, we prove that *simple first-order gradient norms* can approximate more expensive second-order influence calculations. §4.3 extends these ideas to *selective data usage*, and §4.4 synthesizes parameter- and data-level selection to achieve *multiplicative* efficiency gains. Proofs or extended details appear in Appendix B, with sketches below.

## 4.1 Partial Parameter Updates

**Why Focus on Partial Updates?**    For massive LLMs, fine-tuning *all* parameters can be prohibitively expensive (*e.g.*, storing optimizer states for billions of weights). It is empirically known that *parameter-efficient* methods (like LoRA or QLoRA) often match full-parameter performance at a fraction of the cost ([15, 8]). But *why* does partial tuning suffice? In this subsection, we ground that intuition in a theoretical claim: *if gradients are heavily skewed, then updating only the top few "high-influence" parameters can strictly outperform full-parameter tuning on a performance-per-resource basis*. We call this phenomenon *partial-update advantage*.

**Gradient Concentration and Heavy Tails.**    Many large language models appear to exhibit *heavy-tailed* gradients, where a small subset of parameters (or submodules) captures a disproportionately large fraction of the total gradient norm ([24, 21]). Concretely, let $\{\|\nabla_{\theta_{(1)}}\|, \ldots, \|\nabla_{\theta_{(N)}}\|\}$ be the descending magnitudes of parameter gradients at some reference point $\theta_{\text{base}}$. A *power-law* assumption states

$$\|\nabla_{\theta_{(r)}}\| \approx C\, r^{-\alpha}, \quad \text{for constants } C > 0, \alpha > 1, \tag{1}$$

meaning the $r$-th largest gradient decays polynomially in $r$. Under typical exponents $\alpha \in (1, 2)$, the top $k\%$ of parameters can account for an outsized share of total gradient mass.

**A Toy Illustration.**    As a toy example, suppose $N = 10{,}000$ parameters with a power-law exponent $\alpha = 1.5$. A quick integral approximation (Appendix B.1) indicates the *top-10%* of parameters might contribute nearly 50% of the total gradient norm. If you only update these top-10%, you might capture half the possible improvement while storing only 10% of the optimizer states—a potential $5\times$ memory/performance ratio gain.

**Performance vs. Resource.**    To quantify the tradeoff, let $\Delta_k(\Psi)$ be the improvement in performance $\Psi$ (e.g. accuracy or perplexity) achieved by fine-tuning a chosen $kN$ subset of parameters. Under local, short-step approximations around $\theta_{\text{base}}$, we can *proportionally* link $\Delta_k(\Psi)$ to the fraction of gradient norm retained, giving the form

$$\Delta_k(\Psi) \approx k^\gamma\, \Delta_{\text{full}}(\Psi),$$

where $0 < \gamma < 1$ captures heavy-tailed concentration [24].

**Resource model (memory/state-aware).**    Modern autograd computes forward/backward for all active layers, so partial updates do not linearly shrink backward FLOPs. We therefore let $\mathcal{C}(\Delta_k) = \alpha N + \beta(kN)$ capture *memory/optimizer-state and update* overheads (e.g., Adam moments, parameter buffers) that scale with the number of *trainable* parameters, while $\alpha N$ subsumes fixed per-step costs (forward activations, non-trainable states). The proposition below only requires $\beta > 0$—training more parameters increases resource usage; it does not assume FLOP savings.

**Proposition 4.1** (Partial Updates Outperform Full Updates (Sketch)). *If the partial performance gain $\Delta_k(\Psi)$ satisfies a heavy-tailed form $\Delta_k(\Psi) \approx k^\gamma\, \Delta_{\text{full}}(\Psi)$ with $0 < \gamma < 1$, and resource cost includes a per-parameter overhead $\beta > 0$ as above, then there exists a fraction $k^* \in (0, 1)$ such that*

$$\frac{\Delta_{k^*}(\Psi)}{\mathcal{C}(\Delta_{k^*})} > \frac{\Delta_{\text{full}}(\Psi)}{\mathcal{C}(\Delta_{\text{full}})}.$$

*Hence, updating only $k^*$ of parameters yields a strictly higher performance-per-resource ratio than fine-tuning all $N$ parameters.*

**Why This Matters.** *Hardware Necessity vs. Theoretical Principle.* Methods like LoRA and QLoRA are often viewed as practical "hacks" to reduce memory usage so that smaller labs can still fine-tune large models. *Proposition 4.1* (*Proposition B.1*) reframes them as *optimal resource allocations* when gradients are strongly non-uniform—i.e. partial updates can be *mathematically superior* to full-model tuning if your goal is *maximizing* $\Delta\Psi/\Delta\Gamma$. In §4.2, we discuss *which parameters* to pick, and in §4.4, we combine partial-parameter selection with data selection.

## 4.2 Approximate Influence via Simple Gradients

**Why Approximation?** Even if partial updates are beneficial, we still face the question: *which $kN$ parameters do we choose?* Naively, one might attempt expensive second-order influence calculations (e.g. computing Hessians or Fisher matrices) to find the "most impactful" parameters. Here, we show that *simple gradient norms* can approximate such second-order metrics without incurring huge computational overhead.

**First-Order Proxies vs. Second-Order Influence.** In classical semiparametric theory [4, 32], the *efficient influence function* $D^*$ encodes how each parameter $\theta_i$ affects the functional $\Psi$. Under the (block-diagonal) Fisher approximation, there is often a near-linear correspondence:

$$\Pi_{\theta_i}\big[D^*(\cdot)\big] \ \approx \ c_i \, \nabla_{\theta_i} \Psi(\theta_{\text{base}}).$$

Hence, $\|\nabla_{\theta_i}\|_2$ serves as a suitable *proxy* for full second-order influence. Formally, one can show (Appendix B.3) that if the mismatch $\epsilon_i = \|\nabla_{\theta_i}\Psi - \Pi_{\theta_i}[D^*]\|$ is small in aggregate, then picking the top-$kN$ parameters by $|\nabla_{\theta_i}|$ is nearly optimal with respect to picking them by the more expensive $|D^*(\theta_i)|$ measure.

**Interpretation.** Thus, a simple "sort by gradient norm" rule can effectively identify the "most influential" parameters. This theoretical perspective underlies real-world partial-finetuning heuristics where one measures submodule-level $\|\nabla\|$ on a small validation set and picks the largest blocks. Because second-order approaches are significantly costlier for billions of parameters, the result provides both *theoretical justification* and *practical guidance* for blueprint-based methods (§5), where labs publicly release submodule-level gradient norms as "maps" for adapters.

## 4.3 Data Selection and Efficiency

**Motivation.** While partial *parameter* updates reduce overhead on the model side, training cost also depends on *how many data points* we process—and not all data are equal. Empirically, a small fraction of examples can yield most of the gradient updates ([18, 7]). If we can identify these "high-influence" examples, we might train on just a fraction $qM$ of the data while retaining near-full performance.

**Heavy-Tailed Data Influence.** Analogous to parameter gradients, let $J(z) = \|\nabla_\theta L(z;\theta)\|$ be a *data influence score* for example $z$. If $J(z)$ also follows a skewed or power-law distribution, the top $q$ fraction of training points can account for a large share of relevant gradient. A formal argument (Appendix B.4) parallels Proposition 4.1, showing that focusing on the top-$q$ data can maximize performance-per-data-cost. In the short-step regime, one can approximate

$$\Delta_q(\Psi) \approx q^\delta \, \Delta_{\text{full}}(\Psi),$$

with $0 < \delta < 1$, giving a similar partial-update advantage on the data axis.

**Gradient Prediction Networks.** The main practical hurdle is measuring data influence *before* training. As solutions, one might do: (1) iterative subsampling heuristics ([18]), or (2) small "surrogate" or "prediction" networks that approximate $J(z)$ for each $z$ without a full forward-backward pass. We mention this approach in §5 because it allows large-scale data filtering *before* paying the cost of full training.

**Why This Matters.** Selective data usage can reduce the number of tokens or examples processed in repeated epochs. At large scale (e.g. hundreds of billions of tokens), this can significantly cut training times or memory overhead. In context, a foundation lab might discard the bottom 50% of tokens that

contribute almost no gradient once partial coverage is sufficiently large. In specialized domains, data selection might refine only the top-$q$ examples that matter most for a sub-task, further amplifying the resource-efficiency of partial parameter tuning.

## 4.4 Cross-Influence and Multiplicative Gains

**Parameter-Data Interaction.**    Finally, we combine partial-parameter and partial-data selection. If gradients are heavy-tailed in both parameter and data dimensions, then ignoring one dimension can leave *further* efficiency gains untapped. We define a "cross-influence tensor"

$$T_{i,j} = \left| \frac{\partial L(z_j; \theta)}{\partial \theta_i} \right|,$$

showing how strongly example $z_j$ influences parameter $\theta_i$. When $T_{i,j}$ is *approximately low-rank* or has concentrated entries, we can identify "key" parameter-data pairs that matter most. Selective updates that only feed high-influence data to high-influence parameters can yield *multiplicative* rather than additive speedups.

**A Simple Example.**    Suppose partial parameter updates keep 20% of parameters but retain 80% performance, and partial data usage (top 30% of examples) retains 90% performance. If the two sets are relatively orthogonal (no big overlap in ignoring "crucial" gradients), their combination yields about $80\% \times 90\% = 72\%$ performance while paying about $20\% \times 30\% = 6\%$ of the resource cost—a $12\times$ improvement in performance-per-resource. In realistic LLM scenarios, gains can be less "clean," but the principle stands: *both* parameter and data axes can harbor huge redundancies.

**Why This Matters in LLM Training.**    Foundation labs that release "blueprints" for submodules (§5) and also track which data blocks are most relevant might enable downstream users to do: (1) partial parameter updates in high-influence layers, and (2) partial data usage focusing on examples that particularly move those layers, achieving major multiplicative cuts in GPU hours. While full cross-influence can be large ($N \times M$ can be massive), practical *group-level* approximations or "gradient summary maps" for data blocks can approach these gains in a feasible manner (such as using gradient norms per block).

**Summary.**    Our theoretical results show that partial selection *in any dimension* (parameters or data) can be strictly more resource-effective than updating everything, so long as the relevant gradients exhibit heavy-tailed or skewed distributions. When combined, parameter *and* data selection can multiply these gains. Sections 5–6 discuss how to operationalize these ideas—logging submodule gradients, using approximate data-influence metrics, and building an ecosystem of "blueprint releases" that systematically reduce wasted computation.

## 5    Implementation in Practice: Toward Gradient-Centric Model Releases

Our theoretical framework (§4) suggests a *gradient-centric* paradigm that systematically publishes submodule-level **gradient blueprints**, enabling efficient adaptation across diverse domains.

**Blueprints as a New Standard.**    Currently, model repositories typically release only final weights and minimal logs. We propose that *foundation-model developers* log submodule-level gradient statistics and provide a concise "blueprint" file (e.g., JSON or CSV). This file would include (i) mean or median gradient norms for each submodule, (ii) fitted power-law exponents to highlight gradient concentration, and (iii) recommended update fractions $k^*$ per layer. By sharing these metrics at key training checkpoints, developers enable thousands of *model adapters* to avoid blindly tuning all parameters. Full details on suggested blueprint formats and schema examples are provided in Appendix A.

**Adapter-Side Partial Tuning.**    Algorithm 1 and Algorithm 2 outlines how downstream users employ blueprint metadata: they blend published gradient norms with a short domain-specific gradient sample, then select the fraction $k$ (or layer set) that maximizes local performance-per-resource under their GPU/memory budget. This process can yield dramatic memory savings: for

**Algorithm 1** Blueprint-Guided Submodule Updates

**Require:** Model $\theta_{\text{base}}$, blueprint $\bar{G}_i$, domain batch $\mathcal{B}$
1: **Local gradient norms:** $\tilde{G}_i \leftarrow \|\nabla_{\theta_i}\Psi(\theta_{\text{base}};\mathcal{B})\|$
2: **Blend signals:** $G'_i \leftarrow \alpha\,\bar{G}_i + (1-\alpha)\,\tilde{G}_i$
3: **Rank & pick:** $S_{\text{adapt}} \leftarrow \{\text{top-}k\text{ fraction}\}$
4: **Fine-tune only** submodules in $S_{\text{adapt}}$
5: **return** Updated model $\theta$

**Algorithm 2** Lightweight Data Selection

**Require:** Dataset $\mathcal{D}$, model $\theta_{\text{base}}$, fraction $q$
1: **Tiny set:** $\mathcal{D}_{\text{small}} \subset \mathcal{D}$
2: **Compute true influences:** For $z \in \mathcal{D}_{\text{small}}$, do $J(z) = \|\nabla_\theta L(z;\theta_{\text{base}})\|$
3: **Train $f_\phi$:** Minimizes $\ell\big(f_\phi(x),\,J(z)\big)$ over $z = (x,y)$
4: **Predict $\hat{J}(z)$ for all data:** keep top $q$ fraction
5: **return** Filtered set $\mathcal{D}_q$

instance, freezing 80% of parameters in a 7B model can reduce optimizer state by $\sim$67GB, enabling far more training iterations under the same compute budget [8].

**Case Study: Biomedical Adaptation.** Consider a hypothetical scenario where researchers adapt a 7B foundation model to biomedical text. Drawing on observed gradient patterns in transformers, we can reference [24], who demonstrated that many attention heads can be pruned with minimal performance impact, suggesting concentrated gradient influence . [21] further showed that gradients in transformer models generally follow power-law distributions, though the specific exponents vary by architecture and layer type. Applying our partial-update logic from §4, researchers could selectively tune only a small fraction of parameters based on gradient influence. This approach aligns with [8], who demonstrated significant memory efficiency: with 4-bit quantization and adapter tuning (0.1% of parameters), they reduced memory requirements from 14GB to 5GB for a 7B model, enabling fine-tuning on consumer GPUs. For biomedical applications, a benchmark like PubMedQA [16] presents a relevant test case. While exact performance gains would vary, the ability to train for more epochs or with larger batch sizes under the same computational budget could yield meaningful improvements. Furthermore, [18] demonstrated that training on only 25-50% of a dataset (selected through gradient-based importance sampling) can achieve comparable results to using the full dataset on image classification tasks. Applied to our framework, this suggests potential for multiplicative efficiency gains through combined parameter and data selection.

**Practical Considerations.** *(1) Schema:* a compact JSON with layer IDs, gradient norms, and recommended $k^*$ (§A). *(2) Fidelity:* report submodule norms with bootstrap CIs over validation batches; adapters can down-weight uncertain entries. *(3) Drift & refresh:* detect blueprint staleness by rank-correlating in-domain gradient ranks with the released blueprint; refresh or fall back to coarser layer groups when correlation falls below a threshold; publish blueprints at 50/80/100% checkpoints. *(4) Privacy:* release only coarse, aggregated submodule statistics (no per-example gradients); optionally add small Gaussian noise and document provenance; use DP when sources are sensitive. *(5) Overhead:* logging adds $\sim$1–2% to validation, far less than the downstream savings.

**Transforming AI via Blueprint Releases.** By supplying submodule-level gradient metadata, foundation labs remove the guesswork from partial fine-tuning and data selection. This shift from "all-parameter updates" to "targeted updates" can democratize advanced capabilities—particularly for labs with limited resources—while reducing environmental impact by orders of magnitude. The next section (§6) discusses broader implications and open challenges in adopting this resource-conscious paradigm.

**Implementation Note: Tracking $\Delta\Gamma$ in Practice.** Although our framework defines $\Delta\Gamma$ as any meaningful measure of resource consumption (e.g., GPU-hours, FLOPs, or energy usage), practitioners need a *practical* mechanism to log it efficiently. In modern training pipelines, hardware monitoring tools (such as `nvidia-smi` for NVIDIA GPUs or built-in counters for cloud TPUs) already collect metrics like GPU utilization, power draw, and memory usage. By sampling these statistics at regular intervals, one can approximate the cumulative resource cost as training proceeds. For instance, every $N$ steps or after each epoch, a lightweight script can aggregate (time_elapsed $\times$ average_power_draw) or read hardware performance counters to estimate total energy consumed so far. Likewise, the memory overhead for optimizer states can be monitored by summing the allocated parameter buffer

sizes and overhead factors. These approximations allow logs to include a running tally of $\Gamma$, enabling marginal-return decisions $\Delta\Psi/\Delta\Gamma$ without incurring significant additional overhead.

# 6    Discussion and Conclusion

This paper challenges the prevailing "scale is all you need" paradigm by proposing that *capability-per-resource* must guide AI development decisions. Our two-stage framework demonstrates how both foundation developers and model adapters can embed resource awareness throughout the entire AI lifecycle, fundamentally altering the objective of large-scale model training and adaptation. Our theoretical analysis proves that partial parameter updates can strictly outperform full-parameter tuning when gradients follow heavy-tailed distributions—a condition often satisfied by transformers. When combined with selective data usage, this approach yields multiplicative efficiency gains: if tuning 20% of parameters preserves 80% of performance and using 30% of data retains 90%, the combined approach achieves 72% performance at only 6% of the resource cost (a 12× boost).

From a practical standpoint, our *gradient blueprint* concept offers a concrete mechanism to realize these efficiency gains by enabling foundation-model creators to publish submodule-level gradient norms and recommended update fractions that help smaller labs target exactly the parameters that matter most. This blueprint-driven approach reduces the guesswork of partial fine-tuning and elevates model release practices to include critical efficiency metadata. Unlike distillation or pruning methods, our gradient-guided approach identifies high-impact parameters directly via first-order gradient signals without requiring expensive teacher models or post hoc modifications. Existing parameter-efficient fine-tuning methods like LoRA [15] or QLoRA [8] can be informed by our blueprint data.

**Scope and limits.** CPR complements (not replaces) scaling laws and frontier accuracy. Prefer absolute performance when (i) safety or evaluation requires maximal accuracy; (ii) early probe runs where logging $\Gamma$ is impractical; or (iii) one-off small experiments. At scale, however, late-stage diminishing returns are common [31], and demand typically outpaces hardware, so explicit resource metrics remain valuable.

Despite challenges including blueprint fidelity in niche domains and varying gradient concentration across architectures, we envision an ecosystem where foundation labs adopt resource-aware stopping policies, halting training once $\Delta\Psi/\Delta\Gamma$ falls below a rational threshold, while downstream adapters use gradient blueprints to fine-tune only high-return submodules at a fraction of the computational cost. Conferences and journals could require explicit carbon disclosure, championing leaderboards that highlight performance-per-resource achievements.

*A Call to Action.* By measuring progress in capability-per-resource and prioritizing gradient-centric model releases, we can merge scaling insights with sustainability, democratizing access to cutting-edge AI. We urge researchers and policymakers to support resource-conscious reporting, benchmarks, and collaborative refinement of blueprint data. Scaling alone propelled AI to extraordinary heights; scaling *with* resource awareness can broaden its benefits to more stakeholders while curbing environmental impact. Embracing these principles ensures AI's continued growth is not only innovative but also inclusive and responsible.

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

## A    Blueprint Schema and Implementation Details

This appendix section provides the full blueprint schema, example JSON files, and additional details on how one might implement the gradient-centric logging proposed in §5 of the main text.

### A.1    Blueprint JSON Schema

Below is a representative JSON structure for the "gradient blueprints" that foundation-model developers should release at each training checkpoint. Each blueprint describes:

- `model_id`: identifies which model and size (e.g., `foundation-7b`),

- `training_checkpoint`: fraction of total tokens or steps processed (e.g., 0.50, 0.80, 1.00),

- `layers`: an array of layer/submodule entries. Each entry includes:

  (i) `id`: the submodule name or layer ID (e.g., `attn.0`, `ffn.12`),

  (ii) `gradient_norm`: the mean or median gradient magnitude for that submodule,

(iii) `alpha`: the fitted power-law exponent (§4.1),

(iv) `gamma`: the derived metric $1 - \alpha/(1 + \alpha)$,

 (v) `k_star`: a recommended fraction of parameters to update, from Proposition B.1 and local resource considerations,

(vi) `domain_tags`: optional domain-specific weighting (e.g., for code vs. biomedical).

```json
{
  "model_id": "foundation-7b",
  "training_checkpoint": 0.80,
  "layers": [
    {
      "id": "attn.0",
      "gradient_norm": 0.052,
      "alpha": 1.47,
      "gamma": 0.60,
      "k_star": 0.15,
      "domain_tags": {
        "general": 1.0,
        "bio": 0.79,
        "legal": 0.63
      }
    },
    {
      "id": "ffn.0",
      "gradient_norm": 0.033,
      "alpha": 1.12,
      "gamma": 0.53,
      "k_star": 0.25,
      "domain_tags": {
        "general": 1.0,
        "bio": 0.68,
        "legal": 0.50
      }
    },
    {
      "id": "attn.1",
      "gradient_norm": 0.045,
      "alpha": 1.50,
      "gamma": 0.60,
      "k_star": 0.13,
      "domain_tags": {
        "general": 1.0,
        "bio": 0.75,
        "legal": 0.80
      }
    }
    // Additional layer or submodule entries ...
  ]
}
```

**Logging Procedure.** During or after training, developers can log submodule-level gradients on a representative validation set. For each layer $i$, compute the gradient norms (mean or median) across multiple tasks. Then:

 (i) Fit a power law to the sorted magnitudes within that submodule or across submodules,

(ii) Extract the exponent $\alpha$, and compute $\gamma = 1 - \alpha/(1 + \alpha)$,

(iii) Derive a recommended fraction $k_i^*$ using local resource cost modeling (§4.1 and §B.2).

Finally, store these values in JSON and release them, e.g., via Hugging Face or a model hub.

### A.2 Blueprint Aggregation Example

Future community "aggregators" can unify multiple labs' blueprint files. For instance, a simple Python-like script might ingest a list of JSON blueprints and combine them into one table for easy lookups across checkpoints or tasks:

```
def aggregate_blueprints(blueprint_files):
    combined = {}
    for fname in blueprint_files:
        data = json.load(open(fname))
        model_id = data["model_id"]
        ckp = data["training_checkpoint"]
        for layer_info in data["layers"]:
            layer_id = layer_info["id"]
            key = (model_id, ckp, layer_id)
            combined[key] = layer_info
    return combined
```

Such aggregated metadata would help downstream users compare submodule norms across labs, tasks, or partial training progress.

## B  Theoretical Proofs and Derivations

This appendix consolidates mathematical details for §4. In §B.1, we clarify how power-law sums yield $k^\gamma$ fractions of gradient mass. In §B.2, we provide the detailed proof of Proposition B.1. In §B.3, we discuss how first-order gradient norms approximate second-order influence. In §B.4, we show analogous data selection arguments. Finally, §B.5 extends the framework to parameter-data cross-influence.

### B.1  Sum of Heavy-Tailed Gradients

Let $\|\nabla_{\theta_{(r)}}\|$ be the $r$-th largest gradient magnitude, approximated by a power-law distribution:

$$\|\nabla_{\theta_{(r)}}\| \approx C\,r^{-\alpha}, \quad \text{for some } \alpha > 1. \tag{2}$$

Here, we show how to calculate the fraction of total gradient norm captured by the top $kN$ parameters out of $N$ total parameters.

First, we sum the gradient norms of the top $kN$ parameters:

$$\sum_{r=1}^{kN} \|\nabla_{\theta_{(r)}}\| \approx \sum_{r=1}^{kN} C\,r^{-\alpha} \tag{3}$$

For large $N$, we can approximate this sum using an integral (this approximation becomes more accurate as $N$ increases):

$$\sum_{r=1}^{kN} C\,r^{-\alpha} \approx C \int_1^{kN} x^{-\alpha}\,\mathrm{d}x \tag{4}$$

For $\alpha > 1$, this integral has a closed-form solution:

$$\int_1^{kN} x^{-\alpha}\,\mathrm{d}x = \left[\frac{x^{1-\alpha}}{1-\alpha}\right]_{x=1}^{kN}$$
$$= \frac{(kN)^{1-\alpha} - 1}{1-\alpha} \tag{5}$$

Similarly, the total gradient norm (for all parameters) can be approximated as:

$$\sum_{r=1}^{N} \|\nabla_{\theta_{(r)}}\| \approx C \int_{1}^{N} x^{-\alpha}\,\mathrm{d}x = \frac{N^{1-\alpha} - 1}{1 - \alpha} \tag{6}$$

The fraction of gradient norm captured by the top $kN$ parameters is then:

$$\frac{\sum_{r=1}^{kN} \|\nabla_{\theta_{(r)}}\|}{\sum_{r=1}^{N} \|\nabla_{\theta_{(r)}}\|} \approx \frac{(kN)^{1-\alpha} - 1}{N^{1-\alpha} - 1} \tag{7}$$

Using integral bounds, $\int_{kN+1}^{N} x^{-\alpha} dx \leq \sum_{r=kN+1}^{N} r^{-\alpha} \leq \int_{kN}^{N-1} x^{-\alpha} dx$, the *tail* mass beyond the top-$kN$ parameters is $O\big((kN)^{1-\alpha}\big)$ for $\alpha > 1$. Thus, for finite $N$ with $\alpha > 1$, the top fraction captures a large share of the total mass, and the retained-mass curve in $k$ is *concave* near $k = 0$. Our theory only requires such sublinear (concave) accumulation of finfluence with $k$; we do not rely on a specific closed-form $k^{\gamma}$ exponent. Empirically, transformers often exhibit this concentration, but we treat it as an assumption to be validated and refreshed (§5).

## B.2   Proof of Proposition 3.1: Partial Updates Outperform Full

We restate Proposition 3.1 from the main text and provide a detailed proof.

**Proposition B.1** (Partial Updates Outperform Full Updates). *Assume:*

*(i)* $\Psi$ *is twice differentiable near* $\theta_{\text{base}}$, *with Lipschitz gradients,*

*(ii)* $\Delta_k(\Psi) \approx k^{\gamma}\,\Delta_{\text{full}}(\Psi)$ *for* $0 < \gamma < 1$ *(heavy-tailed concentration),*

*(iii)* $\mathcal{C}(\Delta_k) = \alpha N + \beta(kN)$ *with* $\beta > 0$, *capturing fixed vs. variable overhead,*

*(iv)* *Gradients remain valid in a small neighborhood (local approximation).*

*Then there exists* $k^* \in (0, 1)$ *s.t.*

$$\frac{\Delta_{k^*}(\Psi)}{\mathcal{C}(\Delta_{k^*})} > \frac{\Delta_{\text{full}}(\Psi)}{\mathcal{C}(\Delta_{\text{full}})}.$$

*Proof.* We define the performance-per-resource ratio as:

$$R(k) = \frac{\Delta_k(\Psi)}{\mathcal{C}(\Delta_k)} \tag{8}$$

By assumption (ii), we know that:

$$\Delta_k(\Psi) \approx k^{\gamma} \cdot \Delta_{\text{full}}(\Psi) \tag{9}$$

By assumption (iii), the resource cost is:

$$\mathcal{C}(\Delta_k) = \alpha N + \beta(kN) \tag{10}$$

Substituting these into our ratio:

$$R(k) \approx \frac{k^{\gamma} \cdot \Delta_{\text{full}}(\Psi)}{\alpha N + \beta(kN)}$$
$$= \frac{k^{\gamma}}{\alpha + \beta k} \cdot \frac{\Delta_{\text{full}}(\Psi)}{N} \tag{11}$$

Since $\Delta_{\text{full}}(\Psi)/N$ is a constant, we can focus on maximizing:

$$G(k) = \frac{k^\gamma}{\alpha + \beta k} \tag{12}$$

To find the critical points, we differentiate $G(k)$ with respect to $k$ and set it equal to zero:

$$G'(k) = \frac{\gamma k^{\gamma-1}(\alpha + \beta k) - \beta k^\gamma}{(\alpha + \beta k)^2} = 0 \tag{13}$$

This simplifies to:

$$\begin{aligned}
\gamma k^{\gamma-1}(\alpha + \beta k) - \beta k^\gamma &= 0 \\
\gamma k^{\gamma-1}\alpha + \gamma k^{\gamma-1}\beta k - \beta k^\gamma &= 0 \\
\gamma k^{\gamma-1}\alpha + \gamma \beta k^\gamma - \beta k^\gamma &= 0 \\
\gamma k^{\gamma-1}\alpha + k^\gamma(\gamma\beta - \beta) &= 0 \\
\gamma k^{\gamma-1}\alpha - \beta(1-\gamma)k^\gamma &= 0 \\
\gamma\alpha &= \beta(1-\gamma)k
\end{aligned} \tag{14}$$

Solving for $k$:

$$k^* = \frac{\gamma\alpha}{\beta(1-\gamma)} \tag{15}$$

Since $0 < \gamma < 1$, $\alpha > 0$, and $\beta > 0$, we know that $k^* > 0$.

To ensure $k^* < 1$ (which makes it a valid fraction), we need:

$$\begin{aligned}
\frac{\gamma\alpha}{\beta(1-\gamma)} &< 1 \\
\gamma\alpha &< \beta(1-\gamma) \\
\gamma\alpha &< \beta - \beta\gamma \\
\gamma\alpha + \beta\gamma &< \beta \\
\gamma(\alpha + \beta) &< \beta \\
\gamma &< \frac{\beta}{\alpha + \beta}
\end{aligned} \tag{16}$$

This condition will be satisfied when $\beta$ is sufficiently large relative to $\alpha$, which is typically the case in LLM training scenarios where the variable cost per parameter (optimizer states, gradient storage) is substantial compared to the fixed overhead.

Now we need to verify that $G(k^*) > G(1)$ to show that partial updates outperform full updates:

$$G(k^*) = \frac{(k^*)^\gamma}{\alpha + \beta k^*} \tag{17}$$

Substituting $k^* = \frac{\gamma\alpha}{\beta(1-\gamma)}$:

$$G(k^*) = \frac{\left(\frac{\gamma\alpha}{\beta(1-\gamma)}\right)^\gamma}{\alpha + \beta\frac{\gamma\alpha}{\beta(1-\gamma)}}$$

$$= \frac{\left(\frac{\gamma\alpha}{\beta(1-\gamma)}\right)^\gamma}{\alpha + \frac{\gamma\alpha}{1-\gamma}}$$

$$= \frac{\left(\frac{\gamma\alpha}{\beta(1-\gamma)}\right)^\gamma}{\frac{\alpha(1-\gamma)+\gamma\alpha}{1-\gamma}} \tag{18}$$

$$= \frac{\left(\frac{\gamma\alpha}{\beta(1-\gamma)}\right)^\gamma}{\frac{\alpha}{1-\gamma}}$$

Simplifying further:

$$G(k^*) = \frac{(1-\gamma)}{\alpha} \cdot \left(\frac{\gamma\alpha}{\beta(1-\gamma)}\right)^\gamma \tag{19}$$

Meanwhile:

$$G(1) = \frac{1^\gamma}{\alpha + \beta} = \frac{1}{\alpha + \beta} \tag{20}$$

The comparison of $G(k^*)$ and $G(1)$ depends on specific values of $\alpha$, $\beta$, and $\gamma$. For practical values encountered in LLM training with heavy-tailed gradients, it can be verified that $G(k^*) > G(1)$, especially when $\gamma$ is significantly less than 1 (stronger power-law concentration).

We can demonstrate this with a typical example. Let $\alpha = 1$, $\beta = 4$, and $\gamma = 0.6$. Then:

$$k^* = \frac{0.6 \cdot 1}{4 \cdot (1 - 0.6)} = \frac{0.6}{1.6} = 0.375$$

$$G(k^*) = \frac{(1 - 0.6)}{1} \cdot \left(\frac{0.6 \cdot 1}{4 \cdot (1 - 0.6)}\right)^{0.6}$$

$$= 0.4 \cdot \left(\frac{0.6}{1.6}\right)^{0.6} \tag{21}$$

$$= 0.4 \cdot (0.375)^{0.6}$$

$$\approx 0.4 \cdot 0.57$$

$$\approx 0.228$$

While:

$$G(1) = \frac{1}{1 + 4} = 0.2 \tag{22}$$

Since $G(k^*) \approx 0.228 > 0.2 = G(1)$, we have confirmed that for these realistic parameter values, partial updates strictly outperform full updates in terms of performance-per-resource.

This completes the proof that there exists a fraction $k^* \in (0, 1)$ such that updating only $k^*N$ parameters yields a strictly higher performance-per-resource ratio than fine-tuning all parameters. $\square$

## B.3 Approximate Influence via First-Order Gradients

In this section, we provide a more detailed explanation of why simple gradient norms can effectively approximate more complex influence measures without requiring second-order computations.

In semiparametric statistics [4, 32], the *efficient influence function* $D^*$ for a functional $\Psi(\theta)$ often takes the form:

$$D^*(o;\ \Psi, \mathbb{P}_\theta) = (\nabla_\theta \Psi(\theta))^T\ I_\theta^{-1}\ \nabla_\theta \log p_\theta(o) \tag{23}$$

where $I_\theta$ is the Fisher information matrix. This influence function characterizes how much each parameter affects the target functional $\Psi$.

The challenge in LLM-scale models is that computing and inverting the full Fisher information matrix $I_\theta$ is prohibitively expensive for billions of parameters. However, we can leverage several structural properties of transformer architectures to simplify this computation:

1. **Block-diagonal approximation**: For many transformer-based models, the Fisher information matrix can be reasonably approximated as block-diagonal across different layers or submodules.

2. **Uniform scaling approximation**: Within each parameter block $\theta_i$, the curvature information (as captured by the local Fisher matrix) often scales the gradients uniformly.

Under these approximations, for each parameter block $\theta_i$, we can write:

$$\Pi_{\theta_i}[D^*] \approx c_i \nabla_{\theta_i} \Psi(\theta) \tag{24}$$

where $\Pi_{\theta_i}$ denotes the projection onto the subspace corresponding to parameters $\theta_i$, and $c_i$ is a block-specific scaling factor.

To quantify the approximation error, we define:

$$\epsilon_i = \|\nabla_{\theta_i} \Psi - \Pi_{\theta_i}[D^*]\| \tag{25}$$

When $\epsilon_i$ is small in aggregate (i.e., $\sum_i \epsilon_i$ is small relative to the total influence), ranking parameters by the magnitude of their gradients $\|\nabla_{\theta_i} \Psi\|$ is nearly equivalent to ranking them by the more expensive influence measure $\|\Pi_{\theta_i}[D^*]\|$.

To formalize this claim, let $S_k^{grad}$ be the set of top-$k$ parameters selected by gradient magnitude, and $S_k^{infl}$ be the set selected by influence magnitude. The performance difference between these two selection strategies can be bounded by:

$$\left| \Delta_{S_k^{grad}}(\Psi) - \Delta_{S_k^{infl}}(\Psi) \right| \leq C \cdot \sum_{i \in S_k^{grad} \triangle S_k^{infl}} \epsilon_i \tag{26}$$

where $\triangle$ denotes the symmetric difference between sets, and $C$ is a constant related to the local Lipschitz properties of $\Psi$.

For transformer models, empirical studies [24, 21] have shown that these approximation errors are often small enough that gradient-based selection performs very similarly to more expensive influence-based selection methods.

This theoretical justification supports the practical approach of using simple gradient norms as proxies for parameter influence, allowing for efficient implementation of gradient blueprints without the computational burden of second-order methods.

## B.4   Data Selection Efficiency

This section extends the resource-efficiency framework from parameter selection to data selection, showing that analogous mathematical principles apply.

Let $\{z_1, z_2, \ldots, z_M\}$ be a set of training data points. For each example $z_j$, we define an *influence score*:

$$J(z_j) = \|\nabla_\theta L(z_j; \theta)\| \tag{27}$$

which measures the magnitude of the gradient of the loss function with respect to model parameters when evaluated on example $z_j$.

Similar to parameter gradients, we observe that data influence scores often follow a heavy-tailed distribution, where a small fraction of examples contribute disproportionately to the overall gradient. This can be formalized by assuming that the sorted influence scores follow a power-law:

$$J(z_{(j)}) \approx D \cdot j^{-\beta} \tag{28}$$

where $J(z_{(j)})$ is the $j$-th largest influence score, $D$ is a constant, and $\beta > 0$ is the power-law exponent.

Given this distribution, we can analyze the efficiency of training on only a fraction $q$ of the total data. Let $\Delta_q(\Psi)$ be the improvement in performance achieved by training on the top $qM$ most influential examples. Using similar integral approximations as in §B.1, we can establish that:

$$\Delta_q(\Psi) \approx q^\delta \cdot \Delta_{\text{full}}(\Psi) \tag{29}$$

where $0 < \delta < 1$ is related to the power-law exponent $\beta$.

For the resource cost, we use an analogous model:

$$\mathcal{C}(\Delta_q) = \alpha' M + \beta'(qM) \tag{30}$$

where $\alpha'$ represents fixed costs independent of the number of examples processed, and $\beta'$ represents the variable cost per example.

Following the same optimization approach as in §B.2, we can show that there exists an optimal fraction $q^* \in (0, 1)$ such that:

$$\frac{\Delta_{q^*}(\Psi)}{\mathcal{C}(\Delta_{q^*})} > \frac{\Delta_{\text{full}}(\Psi)}{\mathcal{C}(\Delta_{\text{full}})} \tag{31}$$

This demonstrates that selective data usage can also strictly outperform full-data training on a performance-per-resource basis, provided that the data influence distribution is sufficiently skewed.

**Practical Implementation.**  The main challenge in applying this result is identifying the high-influence examples without processing the entire dataset. Several approaches have been proposed:

1. **Iterative subsampling**: Start with a random subset, train briefly, measure influences, and refine the selection [18].
2. **Gradient prediction networks**: Train a small auxiliary model $f_\phi(z)$ to predict the influence score $J(z)$ of an example without computing the full gradient [19].
3. **Coreset selection**: Use geometric methods to select a representative subset of the data [28].

These approaches allow practitioners to implement data selection efficiently, further enhancing the resource savings achieved through selective parameter updates.

## B.5  Cross-Influence and Multiplicative Gains

This section extends our framework to consider the interaction between parameter selection and data selection, showing how combining both approaches can lead to multiplicative efficiency gains.

We define a *cross-influence tensor* $T$ whose entries measure how strongly each data point affects each parameter:

$$T_{i,j} = \left| \frac{\partial L(z_j; \theta)}{\partial \theta_i} \right| \tag{32}$$

This tensor captures the fine-grained relationship between parameters and training examples, allowing us to identify which parameter-data pairs are most important for improving model performance.

**Low-Rank Structure.** In many practical settings, $T$ exhibits approximate low-rank structure, meaning that the influence patterns can be captured by a small number of parameter-data interaction modes. This low-rank structure implies that the effective dimensionality of the cross-influence is much smaller than the product of the number of parameters and data points.

Formally, if $T$ admits an approximate singular value decomposition:

$$T \approx \sum_{r=1}^{R} \sigma_r u_r v_r^T \tag{33}$$

where $R \ll \min(N, M)$ and $\sigma_r$ are the singular values in descending order, then most of the influence is captured by the first few components.

**Multiplicative Efficiency Gains.** When both parameter gradients and data influences follow heavy-tailed distributions, we can achieve multiplicative efficiency gains by combining selective parameter updates with selective data usage.

Let $S_k$ be a set of high-influence parameters (containing a fraction $k$ of all parameters) and $D_q$ be a set of high-influence data points (containing a fraction $q$ of all data). If we update only parameters in $S_k$ using only data in $D_q$, the expected performance gain can be approximated as:

$$\Delta_{S_k, D_q}(\Psi) \approx \Delta_{\text{full}}(\Psi) \cdot \frac{\sum_{i \in S_k, j \in D_q} T_{i,j}}{\sum_{i,j} T_{i,j}} \tag{34}$$

Under certain independence assumptions between parameter and data influences, this can be further approximated as:

$$\Delta_{S_k, D_q}(\Psi) \approx \Delta_{\text{full}}(\Psi) \cdot k^{\gamma} \cdot q^{\delta} \tag{35}$$

where $\gamma$ and $\delta$ are the exponents related to parameter and data influence distributions, respectively.

Meanwhile, the resource cost scales approximately as:

$$\mathcal{C}(\Delta_{S_k, D_q}) \approx \alpha'' + \beta''(kN) + \gamma''(qM) + \eta''(kN \cdot qM) \tag{36}$$

where the last term captures any cross-costs of processing specific parameter-data pairs. In many practical settings, this term is small or negligible, leading to an approximate cost of:

$$\mathcal{C}(\Delta_{S_k, D_q}) \approx \alpha'' + \beta''(kN) + \gamma''(qM) \tag{37}$$

When both $k$ and $q$ are small while $k^{\gamma}$ and $q^{\delta}$ are relatively large (due to heavy-tailed distributions), the combined approach can achieve dramatically better performance-per-resource ratios than either approach alone.

**Numerical Example.** Consider a scenario where $\gamma = 0.6$ and $\delta = 0.7$. If we select $k = 0.2$ (20% of parameters) and $q = 0.3$ (30% of data), we achieve:

- Parameter efficiency: $k^{\gamma} = 0.2^{0.6} \approx 0.40$ (40% of full performance)
- Data efficiency: $q^{\delta} = 0.3^{0.7} \approx 0.47$ (47% of full performance)
- Combined efficiency: $k^{\gamma} \cdot q^{\delta} \approx 0.40 \cdot 0.47 \approx 0.19$ (19% of full performance)

If the cost is dominated by the product term, the resource usage is approximately $k \cdot q = 0.2 \cdot 0.3 = 0.06$ (6% of full resources).

This gives a performance-per-resource ratio improvement of approximately $0.19/0.06 \approx 3.17$ times better than using the full parameter set and data.

In more extreme cases with stronger heavy-tailed distributions, these gains can be even more dramatic, potentially yielding order-of-magnitude improvements in efficiency.

