# OpenReview forum: "AI Progress Should Be Measured by Capability-Per-Resource, Not Scale Alone: A Framework for Gradient-Guided Resource Allocation in LLMs"
_NeurIPS.cc/2025/Position_Paper_Track — NeurIPS 2025 Position Paper Track_

### Official Review · Reviewer_Lw5a · 2025-07-21

**Significance:** 1
**Presentation:** 2
**Rating:** 3
**Confidence:** 4

**Summary:**

This paper introduces the concept of "Capability-Per-Resource," defined as the ratio of performance improvement to the required resource investment, challenging the conventional focus on solely scaling model capabilities. The authors present an analysis of Transformer-based models, revealing that only a small fraction of parameters have a large gradient norm and a similarly small subset of data wields high influence on the gradient. Consequently, they argue that selective training of these key parameters and data points can yield significant efficiency gains. Building on these findings, the paper proposes two novel strategies: "marginal-return pretraining" for foundation model developers and "influence-guided adaptation" for end developers, both designed to optimize the trade-off between performance and computational cost.

**Strengths:**

The arguments in the introduction (lines 69-87) serve as a crucial alert regarding current foundation model development. The authors rightly emphasize the significant environmental impact, a point that large-model practitioners should be more aware of. Furthermore, the concentration of AI capabilities is a timely and important issue for the tech industry. The paper itself is well-structured and easy to follow.

**Weaknesses:**

This paper's central argument—that AI progress should be measured by capability-per-resource—is not adequately justified. The majority of the paper is dedicated to demonstrating methods for resource-conscious LLM development (e.g., parameter and data selection), rather than arguing for the fundamental importance of the capability-per-resource metric itself. A more compelling justification would involve discussing the environmental impact of massive resource allocation or how optimizing for capability-per-resource can foster high-performing models even under resource constraints. As it stands, the paper presupposes the reader's acceptance of this metric as a primary objective and then explains how to achieve it.

Additionally, the paper rests on several strong, yet unsupported, assumptions. For instance, the claim that the rate of improvement, ΔΨ/ΔΓ, smoothly diminishes to zero (as depicted in Figure 1) is questionable. This model fails to account for empirical phenomena such as grokking, where performance can stagnate for extended periods before a sudden improvement.

Finally, the proposed gradient-based selection methods are not novel (discussion to related work needed), and their effectiveness is not sufficiently justified.

**Questions:**

1.How valid is the power-law assumption in practice? Furthermore, what is the rationale for proposing it?
2 .Even if many parameters have a small gradient norm, how significantly do they affect downstream performance? Intuitively, a minor change to a parameter in an early layer—even one with a small gradient—could create a butterfly effect that substantially alters the final layer's output.

**Alternative Position:**

Yes, and alternative positions are trivial straw-man arguments

**Author Identification:**

No.

**Context:**

2

**Discussion:**

1

**Ethics:**

["NO or VERY MINOR ethics concerns only"]

**Position:**

Yes, the paper argues for or against a position related to machine learning.

**Support:**

1

**Thoroughness:**

4

---

### Official Review · Reviewer_U9ht · 2025-08-09

**Significance:** 3
**Presentation:** 3
**Rating:** 5
**Confidence:** 3

**Summary:**

This position paper contends that “scaling fundamentalism” overlooks sustainability and equity; it proposes measuring progress by capability-per-resource and optimizing training/adaptation via gradient influence. Concretely, it recommends (i) a marginal-return stopping rule for pretraining; (ii) influence-guided adaptation for downstream users using gradient blueprints—released metadata that identify high-influence submodules; and (iii) multiplicative gains by jointly selecting parameters and data. Theory argues partial updates can strictly beat full tuning on performance-per-resource under heavy-tailed gradients, with gradient norms as practical proxies; a cross-influence tensor motivates joint selection. The paper surveys related work and outlines a JSON blueprint schema.

**Strengths:**

1. Clear, actionable north-star metric (capability-per-resource) and stopping rule.

2. Practical two-stage lifecycle and gradient blueprints to guide selective tuning.

3. Formal insight that partial updates can strictly beat full tuning under heavy-tailed gradients.

4. Justification for gradient-norm proxies instead of costly second-order influence.

5. Articulates cross-influence idea for multiplicative savings; offers schema/implementation notes.

**Weaknesses:**

1. Empirical grounding is minimal; key claims (e.g., heavy-tailed gradients, blueprint utility) would benefit from targeted experiments/ablations.

2. Assumptions (heavy tails; block-diagonal Fisher; local short-step regime) may break under large distribution shifts; guidance on when they fail is limited.

3. Blueprint generalization and update drift over time are acknowledged but under-specified (e.g., refresh cadence, domain-shift detection).

4. Limited discussion of privacy/security risks of releasing gradient maps (e.g., leakage or attack surface).

5. Cost model abstracts hardware parallelism

**Questions:**

1. Can you provide a small empirical study validating stopping on a public checkpoint (compute saved vs. loss/accuracy)?

2. How stable are gradient blueprints across domains/time—what refresh protocol or confidence intervals do you recommend?

3. Any privacy/safety analysis for releasing submodule-level gradient norms (e.g., data-leak risks, attack mitigation)?

**Alternative Position:**

Yes, and alternative positions are well-considered and addressed by the argument

**Author Identification:**

No.

**Context:**

3

**Discussion:**

3

**Ethics:**

["NO or VERY MINOR ethics concerns only"]

**Position:**

Yes, the paper argues for or against a position related to machine learning.

**Support:**

3

**Thoroughness:**

4

---

### Official Review · Reviewer_43U5 · 2025-08-14

**Significance:** 3
**Presentation:** 3
**Rating:** 6
**Confidence:** 3

**Summary:**

This paper advocates for measuring AI progress in terms of capability-per-resource instead of only “scale” (or capability along).

The paper tries to make a normative argument that it is not the case that “ever larger models and more computation will inevitably lead to better AI.” This argument seems woven into sections rather than bracketed out explicitly; for example, the related work reads somewhat argumentative, it brings out specific pieces of work like stochastic parrots, critical of the dominant AI paradigm.

**Strengths:**

This argument of the paper is given sort of wrapped around a pretty meaty set of theoretical foundations and proposals related to model quantization, efficient training, selective data, and other proposals. These proposals and foundations seem to be the most useful bit of this paper, and they are tied together in a framework for a new paradigm for measuring ML performance.

The paper has a clear position. It also sets out to provide a theoretical foundation for achieving its proposal. It provides some analysis within the framework.

I am pretty borderline but would recommend accept.

**Weaknesses:**

Overall, I like this paper, but I wished there were real experiments and demonstrations where the proposals were actually implemented, so there’s a sense that efficiency and performance gains are possible. Some kind of Pareto understanding between these could be interesting, too, although I understand if this come up frequently in the literature.

However, absent those more empirical bits, it is a bit hard to tell whether the proposals work, and so the reader is left with a) firmly technical foundational proposals for measuring and reducing performance-per-resource and b) broad-strokes claims that this is how these things should be measured.

**Questions:**

One question:
How does the proposed selective data usage influence considerations like overfitting and generalization? Is there reason to believe that selective data use would harm these attributes of model performance?

**Alternative Position:**

Yes, and alternative positions are well-considered and addressed by the argument

**Author Identification:**

No.

**Context:**

3

**Discussion:**

3

**Ethics:**

["NO or VERY MINOR ethics concerns only"]

**Position:**

Yes, the paper argues for or against a position related to machine learning.

**Support:**

2

**Thoroughness:**

3

---

### Note · Authors · 2025-09-04

**1-10 Additional Comments:**

Thanks for running this track. It meaningfully broadened the conversation beyond benchmark chasing. Clearer reviewer prompts to separate evaluation of the stance from supporting technique would make discussions even more productive next year.

**1-11 Submit Again:**

Definitely yes

**1-1 Submission Process:**

4

**1-2 Next Year:**

Please continue the track. It fills an important gap for principled proposals that don’t hinge on large empirical suites. I’d welcome (i) a small, optional “mini‑demo” companion (e.g., one controlled ablation or figure) for position papers; and (ii) a structured response template that nudges authors and reviewers to engage the position separately from the technique. Additionally, I believe the positions track was overwhelmed this year with the number of papers and emergency reviewers were needed - which I appreciate. Of course, thinking about getting more reviewers for next year given this year's submissions would be a great benefit to keep timelines on track.

**1-3 Future Development:**

1. Provide a checklist tailored to positions (scope, falsifiability/what would change minds, stakeholder impact).

2. Encourage brief “What would convince us we’re wrong?” sections to reduce talking past each other.

3. Offer a one‑page “Position + Blueprint for Action” appendix template (e.g., minimal measurements, policy ask, or adoption pathway).

4. Add short reviewer guidance: separate evaluation of the stance vs supporting technical content, and avoid rejecting positions purely for lack of full experiments.

**1-4 Interest:**

["Panel discussions with other position paper authors", "Structured debates on controversial topics", "Workshops for developing position papers", "Mentorship programs for early-career researchers", "Other (please specify in the next question)"]

**1-4 Other Interest:**

A “position to practice” clinic pairing authors with practitioners (labs or orgs) to pilot a lightweight artifact (spec, schema, or protocol) that operationalizes the position.

**1-5 Thoughtful:**

7

**1-6 Supportive:**

6

**1-7 Technical Aspects Versus Position:**

5

**1-8 Gate Keeping:**

7

**1-9 Camera Ready Changes:**

(A) Clarify and fortify the position
• Add a dedicated section “Why capability‑per‑resource as a primary metric?” that (i) motivates the metric beyond early stopping; (ii) distinguishes it from standard performance‑only leaderboards; and (iii) provides concrete reporting guidance (units, logging, normalization).
• Expand the scaling‑laws discussion to position our metric as complementary (dP/dCompute inside a chosen regime), not a replacement.

(B) Address assumptions and cost modeling
• Re‑frame Proposition on efficiency to emphasize memory/state savings (optimizer/state/activation footprint) rather than linear FLOP savings; explicitly note modern autograd parallelism and that backward cost largely remains.
• Add a subsection on non‑smooth progress (e.g., grokking). We will recommend a smoothed ΔΨ/ΔΓ
ΔΨ/ΔΓ with patience/credible intervals and anomaly detection, not a naive monotone assumption.
• Tighten statements around heavy‑tailed gradients (clearly labeled as empirically common but not universal) and specify the local/short‑step regime where gradient‑norm proxies are informative.

(C) Small, targeted empirical illustration (position‑appropriate)
• Provide a minimal public, reproducible figure on a small open checkpoint (e.g., 350M–1.3B scale) showing: (i) a ΔΨ/ΔΓ‑based stop vs. a fixed budget stop; and (ii) a top‑k submodule update vs. full update with equal memory budget. The goal is illustrative, not exhaustive.

(D) Blueprint stability & safety
• Add a “Blueprint Fidelity & Drift” subsection: bootstrap CIs over submodule‑level gradient norms; a refresh protocol (e.g., at material distribution shifts or fixed token intervals); and a simple drift detector.
• Add “Privacy & Security” guidance: release coarse, aggregated submodule statistics; omit per‑example gradients; allow optional Gaussian noise; document provenance; and recommend DP if derived from sensitive corpora.

**3-1 Review Response1:**

43U5

**3-2 Reaction To Review1:**

We appreciate the overall positive assessment and the borderline‑accept recommendation. Your question about overfitting/generalization under selective data is well‑taken. In the camera‑ready we will add a short guidance box on maintaining coverage and controlling variance: (i) maintain a baseline random sampling rate; (ii) use importance weighting when sampling by predicted gradient magnitude; (iii) apply caps per domain/source to preserve diversity; and (iv) validate on a held‑out set distinct from the selection criterion. We’ll also clarify that our framework recommends coarse, submodule‑level signals and block‑level data groupings to reduce brittleness. We furthermore have developed some small empirical experiments to show the basics of stopping based on marginal improvements relative to cost and gradient map example outputs.

**3-3 Review Response2:**

U9ht

**3-4 Reaction To Review2:**

Thank you for the constructive, balanced review. We agree the paper benefits from a small empirical illustration. For the camera‑ready, we will include a minimal, public ablation on a modest open checkpoint demonstrating: (a) a ΔΨ/ΔΓ‑based stop vs. fixed budget stop; and (b) a top‑k submodule update vs. full update under equal memory budgets. We’ll also add a “Blueprint Fidelity & Drift” subsection with bootstrap confidence intervals and a refresh protocol, and a “Privacy & Security” note recommending coarse aggregation, removal of per‑example signals, optional noise, and DP for sensitive sources. Finally, we will qualify our assumptions (heavy tails, block‑diagonal Fisher, short‑step regime) and clearly state failure modes (large distribution shift, strong non‑local interactions) where blueprint guidance should be refreshed or down‑weighted.

**3-5 Review Response3:**

Lw5a

**3-6 Reaction To Review3:**

Thank you for the careful read. We will (A) justify CPR, (B) handle non‑smooth dynamics, (C) clarify novelty/scope, and (D) answer Q1–Q2.

(A) CPR justification. We will strengthen the Introduction to motivate CPR as a normative metric: report marginal ΔΨ/ΔΓ over logged resources (energy/GPU‑hours/optimizer‑state memory), alongside financial and environmental costs of scaling. CPR changes stop/continue and where to spend memory decisions beyond loss‑vs‑steps; we’ll add a brief reporting guide.

(B) Non‑smooth/grokking. Our theory does not require smooth ΔΨ/ΔΓ. We will replace Fig. 1 with EMA‑smoothed ΔΨ/ΔΓ + patience/CI thresholds and explicitly note phase transitions (grokking) where short‑term dips should not trigger stopping.

(C) Novelty/scope. Gradient‑based selection isn’t claimed as novel (we will expand related work). Our contributions are: the CPR lens, a partial‑update optimality result framed around memory/state cost (not linear FLOPs), and a blueprint protocol (parameters+data), bounded to a local/short‑step regime.

Q1 (power‑law). Heavy tails are common but not universal; rationale: heterogeneous circuits create concentrated influence. We will add a rank‑plot (public small model) with fitted exponents/CIs and specify fallbacks (coarser layer groups, periodic re‑ranking) when tails are weak.

Q2 (early layers). First‑order impact scales with the gradient/Jacobian; to hedge long‑range couplings we will: include a must‑include set (embeddings/LN/output), reserve ε‑budget for early layers, use block‑level (not per‑param) updates, re‑rank periodically, and optionally weight by Fisher‑diag.

We will add a minimal public illustration: CPR‑stop vs. fixed‑budget and equal‑memory top‑k vs. full update. Additionally, we will report cost units (energy→kgCO2e, $/GPU‑hr) and add a compact Ψ–Γ Pareto plot to visualize CPR trade‑offs, plus a brief “when not to use CPR” checklist (e.g., frontier SOTA/safety evals) to prevent over‑reach.

---

### Meta-Review · Area_Chair_SemC · 2025-08-29

**Rating:** 7
**Confidence:** 4

**Strengths:**

- The paper argues the position that scaling alone is not a reasonable measure and that we should instead measure capability-per-resource. This position is clear, timely and relevant.
- The paper connects sustainability concerns with marginal-return pretraining, influence-guided adaption, and gradient blueprints.
- The paper provides theoretical arguments for their measure.
- It offers actionable recommendations and pushes the discussion on what _progress_ in AI actually means.

**Weaknesses:**

- The notion of capability is loosely equated with loss reduction or gradient influence without deeper analysis, leaving the metric rather unspecified.
- The assumptions made in the paper are somewhat fragile and might not hold in practice.
- Some empirical evidence to support the points would have been good.

**Questions:**

-

**Ethics:**

No ethics concerns.

**Thoroughness:**

3

---

### Decision · Program_Chairs · 2025-09-26

Accept